# Association between birth weight and risk of overweight at adulthood in Labrador dogs

**Amélie Mugnier** [1] *, **Anthony Morin**[2], **Fanny Cellard**[1], **Loïc Devaux**[1], **Magalie Delmas**[3], **Achraf Adib-Lesaux**[4], **John Flanagan**[4], **Jérémy Laxalde**[4], **Sylvie Chastant** [1], **Aurélien Grellet**[1]

**1** NeoCare, Université de Toulouse, ENVT, Toulouse, France, **2** CESECAH, Lieu-dit Monsable, Lezoux, France, **3** Ecole des Chiens Guides d'Aveugles du Grand Sud-Ouest, Toulouse, France, **4** Royal Canin, Aimargues, France

\* amelie.mugnier@envt.fr

**Data Availability Statement:** All relevant data are within the manuscript and its Supporting Information files.

## Abstract

Several studies in humans indicate that low birth weight predisposes individuals to obesity in later life. Despite the constant increase in prevalence of obesity in the canine population and the major health consequences of this affection, few investigations have been carried out on the association between birth weight and the development of overweight in dogs. The purpose of the current study was to examine the association between birth weight and some other neonatal characteristics and overweight at adulthood in a population of purebred Labrador dogs. Information was collected about the sex, age, neuter status, birth weight, and growth rates (between 0–2 days and 2–15 days of age) in 93 Labrador dogs raised under similar environmental conditions until two months old. The body condition scores (BCS, scale of 1–9) of these dogs at adulthood were recorded, with BCS equal to or greater than 6 classified as overweight. Dogs were split into two groups based on the median birth weight in the population: lower than the median (LTM) and higher than the median (HTM). A logistic regression model was applied to analyse associations between the general characteristics of the dogs (sex, age, neuter status), early life parameters (birth weight, growth rates) and overweight at adulthood. Birth weight was the only early-life parameter found to be associated with overweight (p value = 0.032) with a prevalence of overweight of 70% among the dogs with LTM birth weight vs. 47% in dogs born with HTM birth weight. Overweight was also associated with age and neuter status (p value = 0.029 and 0.005 respectively). Our results suggest that, as in humans, dogs with the lowest birth weights are more likely to become overweight at adulthood. More studies are needed to further examine this relationship and to explore the underlying mechanisms. A subsequent objective could be to identify preventive strategies such as an adapted early nutrition programme for at-risk individuals.

## Introduction

Excessive bodyweight (overweight and obesity) is a growing global health problem in dogs, the prevalence ranging between 20–40% all over the world [1–3]. Multiple risk factors have been described in the literature such as breed, genetics, neuter status, the amount of physical activity

**Funding:** This study was partially funded by Royal Canin SAS (Aimargues, France). The funder provided support in the form of salaries for AM. Moreover, AAL, JL and JF participated in the analyses of the data and in the reviewing of the paper. There was no additional external funding received for this study.

**Competing interests:** The authors have declared that no competing interests exist. The commercial affiliation does not interfere with the complete and objective presentation of this study, nor does it alter their adherence to PLOS ONE policies on sharing data and materiels.

and the type of diet [2, 4, 5]. Excessive bodyweight has a negative effect on health, life quality and life expectancy in both dogs and humans [6–8]. It is also known to predispose to or to exacerbate numerous other diseases such as orthopaedic diseases [6, 9], reproductive disorders [6, 10] and cardiorespiratory diseases [16].

Diseases in adulthood are increasingly associated with early life events [11] and numerous studies have demonstrated an association of low birth weight with obesity risk in humans [12–14]. One explanation could be the thrifty phenotype hypothesis first proposed by Hales and Barker [15] which suggests that, when the energy supply is restricted, the offspring develop early-life metabolic adaptations to promote survival. An association between low birth weight and adult body condition has also been described in pigs [16], mice [17] and guinea pigs [18] but has never been studied in dogs.

Thus, the purpose of this study was to analyse the association between neonatal parameters (birth weight and early growth) and overweight at adulthood in a Labrador dog population.

## Materials and methods

### Study population

**The dog population.** The study was conducted in a guide dog breeding kennel (Centre d'Etude de Sélection et d'Elevage pour Chiens Guides d'Aveugles et Autres Handicapés, CESE-CAH; Lezoux, France), a nonprofit organization affiliated to the French Federation of Guide Dog Associations (FFAC). Only breeding bitches and retired dogs of Labrador breed born at the CESECAH and over one year of age were included. Outside the reproduction period (breeding bitches), or once retired (bitches and sires), the dogs are housed in volunteer families. Breeding bitches included in the reproduction program are housed in the maternity room at the CESECAH's facility, from two weeks before whelping until the separation from the puppies at around 9 weeks after birth. Each puppy is routinely weighed during this period. After weaning, the bitches go back to their volunteer families, and the puppies are transferred to different French Guide Dog Schools, where they are selected and trained to become guide dogs.

**Data collection.** All volunteer families were asked to bring their dogs for an examination session (one exam per dog at adult age in September 2017 or in June 2018). Body fat mass was estimated from the body condition score (BCS). Two operators, using palpation and visual observation, assigned a score to each dog, based on the 9-point scale developed by Laflamme et al. [19]: 1 emaciated, 2 very thin, 3 thin, 4 underweight, 5 ideal, 6 overweight, 7 heavy, 8 obese and 9 grossly obese. At the same time, the age, sex and neuter status were recorded. Finally, information about the neonatal period (birth weight, growth rate between birth and Day 2 and between Day 2 and Day 15) was collected from the CESECAH database.

### Statistical analysis

The interplay between neonatal parameters and the risk of overweight was investigated by fitting a logistic regression which took the general parameters into account.

The outcome was a binary variable based on the BCS. Dogs with a BCS of 5 or less were classed as "not overweight" and those with a BCS of 6 or more, as "overweight". The reference category was dogs that were not overweight. Explanatory variables included sex (male/female), age (continuous, in years), neuter status (neutered/entire), birth weight, growth rate 0–2 days [(weight at 2 days–weight at birth) ÷ weight at birth x 100] and growth rate 2–15 days [(weight at 15 days–weight at 2 days) ÷ weight at 2 days x 100]. The three last parameters were categorised into two groups based on the median (LTM and HTM for lower than, and higher or equal to, the median, respectively).

Starting with the full model (*BCS category ~ Birth weight + Growth rate 2d + Growth rate 15d + Neuter status + Sex + Age*), a backward selection based on the Akaike's Information Criterion (AIC) was applied to select the most parsimonious model. At the end of the backward selection, interactions were tested by running models with and without the interaction term. A Chi-square test was then performed to examine the statistical significance of the residuals sum of squares difference between the two models (with vs. without interaction term). The final model was assessed by Pearson residuals test. Moreover, the area under the receiver operating characteristic curve (AUROC) was used to assess the ability of the models to differentiate overweight and non-overweight dogs.

All statistical analyses were performed using R studio, version 3.4.2 [20]. Statistical significance was defined as P value < 0.05. Statistical uncertainty was assessed by calculating the 95% binomial confidence intervals (95% CI).

## Ethics approval

The protocol was reviewed and approved by the Royal Canin Committee for Animal Ethics and Welfare, reference 050917–39.

## Results

### Population

The present study involved 93 Labradors. Ages ranged from 1 to 9.5 years (median: 3.9 years). Sex distribution was 6 (6.5%) entire males, 36 (38.7%) entire females, 18 (19.4%) neutered males and 33 (35.5%) neutered females. The overall prevalence of overweight (BCS $\geq$ 6) was 58.1% (95% confidence interval, 95% CI: 47.4–68.2, Fig 1). The median birth weight was 420 g (interquartile range, IQR: 380–470), the median growth rate at 0–2 days was 8.2% (IQR: 0–13.9) and the median growth rate at 2–15 days was 175.6% (IQR: 150–194.3). The characteristics of the two groups of interest (overweight and not-overweight) are presented in Table 1.

### Risk factors for overweight

The final logistic regression model presented an acceptable discrimination ability with an AUROC of 0.74 (95% CI: 0.57–0.9). The results indicated that neutering, LTM birth weight

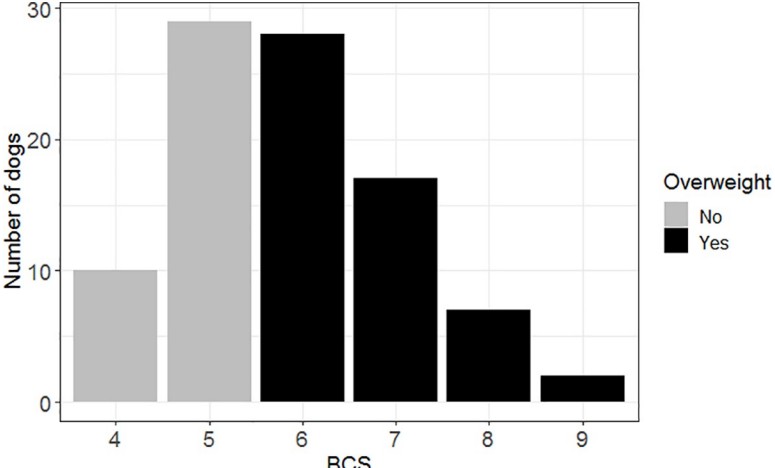

**Fig 1. Body condition score distribution (n = 93).** Body condition score (BCS) was assessed using a 9-point scale [19].

**Table 1. Neonatal and adult characteristics of overweight and not-overweight Labradors (No. = 93).**

| Characteristics | Overweight | Not-overweight |
|---|---|---|
| No. of dogs | 54 | 39 |
| Age (years) | | |
| Median | 4.1 | 2.4 |
| Range | 1–9.6 | 1.1–8.2 |
| Sex [No. (%)] | | |
| Female | 39 (72%) | 30 (77%) |
| Male | 15 (28%) | 9 (23%) |
| Neutered status [No. (%)] | | |
| Entire | 16 (30%) | 26 (67%) |
| Neutered | 38 (70%) | 13 (33%) |
| Birth weight [No. (%)] | | |
| Lower than the median | 32 (59%) | 14 (36%) |
| Higher than the median | 22 (41%) | 25 (64%) |
| Growth rate 0–2 days [No. (%)] | | |
| Lower than the median | 27 (50%) | 19 (49%) |
| Higher than the median | 27 (50%) | 20 (51%) |
| Growth rate 2–15 days [No. (%)] | | |
| Lower than the median | 25 (46%) | 21 (54%) |
| Higher than the median | 29 (54%) | 18 (46%) |

and older age were related to the prevalence of overweight (Table 2, Fig 2). Among the neutered dogs, 75% were overweight vs. 38% in entire dogs (Fig 2A). Among the dogs with LTM birth weight, 70% (95% CI: 54–82) were overweight vs. 47% (95% CI: 32–62) in dogs born with HTM birth weight (Fig 2B). No influence was found of sex, growth rate at 0–2 days or at 2–15 days on the risk of overweight.

## Discussion

The results of the present study in a Labrador dog cohort suggest that birth weight, in addition to age and neuter status, was associated with the risk of becoming overweight at adulthood.

Long-term follow-up in the canine species is difficult, and the population studied here is one of the first for which both neonatal and adult parameters were available. The study was conducted on Labradors raised under similar environmental conditions until two months of age which meant that the breed and breeding conditions were homogeneous. Puppy weights during the first two weeks were prospectively and precisely recorded by CESECAH to ensure high data quality. Body fat mass in the adults was estimated from the BCS. The results of this

**Table 2. Risk factors for overweight: Final logistic regression model (93 Labrador dogs).**

| Parameters | P value | Odds ratio | 95% CI |
|---|---|---|---|
| Age | 0.029 | 1.26 | 1.03–1.58 |
| Birth weight [HTM as Reference] | 0.032 | 2.81 | 1.11–7.47 |
| Growth rate at 0–2 days | Excluded after backward selection | | |
| Growth rate at 2–15 days | Excluded after backward selection | | |
| Neutered status [No as Reference] | 0.005 | 3.83 | 1.51–10.11 |
| Sex | Excluded after backward selection | | |

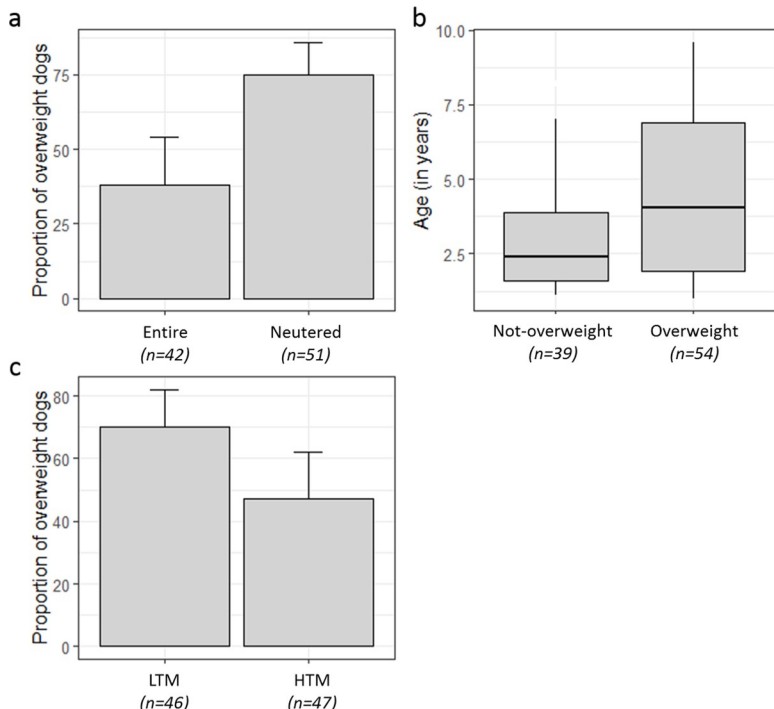

**Fig 2.** Association between overweight and neutered status (a, p = 0.005), age (b, p = 0.029) and birth weight (c, p = 0.032) in 93 Labrador dogs. LTM and HTM for birth weight value lower and higher than the median respectively.

subjective non-invasive method, based on visual assessment and palpation, are well correlated with those obtained with more accurate methods, like dual-energy X-ray absorptiometry and deuterium oxide ($D_2O$) dilution, which are difficult to conduct in the field [19, 21]. Nevertheless, despite this good correlation, further studies with an assessment of adiposity by quantitative measures should be conducted.

We selected our study cohort from a Labrador dog population, because of its well-known predisposition to become overweight compared to other canine breeds [9, 22, 23]. The prevalence of overweight and obesity in our population was 58%, vs 41% in the US Labrador population [3], i.e. slightly higher than the prevalence reported worldwide in multi-breed studies [1–3]. In addition to breed predisposition, this may be explained by the prevalence of neutering, which was also slightly higher in our study (54%) than in the comparative studies (around 40%) [1–3].

As reported elsewhere [3, 24], the prevalence of overweight in our Labrador population increased with age and neutering status. Both conditions (neutering and aging) are associated with an increase in food intake associated with a decrease in energy requirements leading to overweight development [25]. We did not investigate other risk factors for obesity which might have influenced the results, such as the dog's lifestyle [2]. Information regarding recent weight gain or weight loss was not available in adult dogs included, so it was not known if the dogs had recently undergone weight loss programs. Neither was it known if those apparently healthy dogs, except for overweight, were suffering from other conditions that could cause excess weight gain.

As demonstrated in humans [12–14, 26], guinea pigs [18] and rats [27], birth weight was found to be associated with overweight status at adulthood, with a higher prevalence of overweight when the birth weight was below the median in our population. Low birth weight

piglets produce lower quality carcasses with a higher intramuscular fat percentage, than piglets that are heavier at birth [28, 29]. The precise mechanisms responsible for this association between impaired foetal development (leading to a low birth weight) and overweight in later life remain unclear and fall within the overall concepts of Developmental Origin of Health and Diseases (DoHad) and "foetal programming". The latter describes the process whereby particular events occurring during early life have permanent effects on subsequent physiology and metabolism [30]. Hales and Barker [15] hypothesized that intrauterine growth retardation, leading to low birth weight, promotes the development of a "thrifty phenotype" leading to metabolic changes that increase nutritional efficiency and predispose to overweight in later life. Studies in humans and with animal models suggest that this predisposition may be linked to the development of insulin resistance [12], adipocyte hyperplasia [31], alteration of plasma leptin concentration [32, 33], and upregulation of the adipogenic signalling cascade, leading to an increased susceptibility to retain fat in the adipocytes and thus an increased propensity for adiposity [34]. Impaired glucose tolerance [14], modulation of the programming of appetite-regulating hormones with an increased in plasma ghrelin concentration, a significant appetite stimulator secreted by the stomach, might also be involved [27]. Although the present study provides original and novel findings in the canine species, the results cannot be generalized due to the relatively small sample size. A broader set of data including more Labrador dogs and dogs from other breeds should be collected to validate the current findings. Moreover, further studies are needed to explore and identify the precise mechanisms explaining the association between low birth weight and adult overweight in the canine species.

In contrast to previous studies in dogs and humans [35, 36], early growth rates (0–2 days and 2–15 days) were not found associated with the risk of overweight in the present study. This result supports the safety of an early energy supplementation for low birth weight puppies, known to reduce neonatal mortality in puppies [37].

Prospective birth cohort studies with life-long follow-up would help to more accurately explore early-life predictive factors for adult overweight and obesity in the canine species and to quantify the relative impact of early life risk factors as well as environmental factors.

## Conclusion

This study suggests that low birth weight, in addition to its influence on neonatal mortality, is associated with an increased risk of overweight at adulthood in the canine species, even after adjusting for age and neuter status. These findings could make it easier to identify dogs with an increased risk of becoming overweight when adult, at a very early stage (at birth). Early management of these predisposed puppies could help to reduce the prevalence of overweight in the dog population and thus be beneficial for the health and welfare of companion dogs.

## Supporting information

**S1 Dataset.**
(CSV)

## Acknowledgments

We would like to acknowledge the contributions of our team of veterinary students that contributed to the examination sessions. We are also deeply grateful to the staff of the CESECAH and of the Guide Dogs training school of Toulouse for giving their time for animal handling and for providing access to the animals. Finally, we wish to thank the owners who brought in their dogs for the examination sessions and who answered our questions.

## Author Contributions

**Conceptualization:** Amélie Mugnier, Sylvie Chastant, Aurélien Grellet.

**Data curation:** Amélie Mugnier, Aurélien Grellet.

**Formal analysis:** Amélie Mugnier, Achraf Adib-Lesaux, John Flanagan, Jérémy Laxalde, Aurélien Grellet.

**Funding acquisition:** Achraf Adib-Lesaux, John Flanagan.

**Investigation:** Amélie Mugnier, Anthony Morin, Fanny Cellard, Loïc Devaux, Magalie Delmas, Sylvie Chastant, Aurélien Grellet.

**Methodology:** Amélie Mugnier, Achraf Adib-Lesaux, John Flanagan, Jérémy Laxalde, Sylvie Chastant, Aurélien Grellet.

**Project administration:** Amélie Mugnier, Sylvie Chastant, Aurélien Grellet.

**Resources:** Anthony Morin, Magalie Delmas, Sylvie Chastant, Aurélien Grellet.

**Supervision:** Sylvie Chastant, Aurélien Grellet.

**Visualization:** Amélie Mugnier, Fanny Cellard, Loïc Devaux, Sylvie Chastant, Aurélien Grellet.

**Writing – original draft:** Amélie Mugnier, Sylvie Chastant, Aurélien Grellet.

**Writing – review & editing:** Amélie Mugnier, Anthony Morin, Fanny Cellard, Loïc Devaux, Magalie Delmas, Achraf Adib-Lesaux, John Flanagan, Jérémy Laxalde, Sylvie Chastant, Aurélien Grellet.

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
