## [Decision Letter · Decision Letter 0]

10 Nov 2020

PONE-D-20-33064

Association between birth weight and risk of overweight at adulthood in Labrador dogs

PLOS ONE

Dear Dr. Mugnier,

Thank you for submitting your manuscript to PLOS ONE. After careful consideration, we feel that it has merit but does not fully meet PLOS ONE’s publication criteria as it currently stands. Therefore, we invite you to submit a revised version of the manuscript that addresses the points raised during the review process.

Your manuscript was reviewed by two experts in the field, and they have requested some minor changes be made prior to acceptance.

If you could make these modifications and write a brief response to reviewers, that will greatly expedite review upon resubmission.

I wish you the best of luck with your revisions.

Hope you are keeping safe and well in these difficult times.

We look forward to receiving your revised manuscript.

Kind regards,

Simon Clegg, PhD

Academic Editor

PLOS ONE

"This study was partially funded by Royal Canin SAS (Aimargues, France). AAL, JL

and JF participated in the analyses of the data and in the reviewing of the paper."

We note that one or more of the authors have an affiliation to the commercial funders of this research study : Royal Canin.

3.1. Please provide an amended Funding Statement declaring this commercial affiliation, as well as a statement regarding the Role of Funders in your study. If the funding organization did not play a role in the study design, data collection and analysis, decision to publish, or preparation of the manuscript and only provided financial support in the form of authors' salaries and/or research materials, please review your statements relating to the author contributions, and ensure you have specifically and accurately indicated the role(s) that these authors had in your study. You can update author roles in the Author Contributions section of the online submission form.

3.2. Please also provide an updated Competing Interests Statement declaring this commercial affiliation along with any other relevant declarations relating to employment, consultancy, patents, products in development, or marketed products, etc.  

Reviewers' comments:

Reviewer's Responses to Questions

**Comments to the Author**

1. Is the manuscript technically sound, and do the data support the conclusions?

Reviewer #1: Yes

Reviewer #2: Yes

2. Has the statistical analysis been performed appropriately and rigorously? 

Reviewer #1: I Don't Know

Reviewer #2: Yes

3. Have the authors made all data underlying the findings in their manuscript fully available?

Reviewer #1: Yes

Reviewer #2: Yes

4. Is the manuscript presented in an intelligible fashion and written in standard English?

Reviewer #1: Yes

Reviewer #2: Yes

5. Review Comments to the Author

Reviewer #1: This manuscript is about the association between birth weight and overweight in Labrador dogs.

The manuscript is well written, the statistical analysis seems appropriate to me as well as the conclusions, according to the results. However, the work has some limitations and needs some clarification.

1) Since the age of the included dogs ranged from 1 to 9 years of age, it is assumed that this is a retrospective study. Authors are asked to explain how puppies were raised under similar environmental conditions over such a long period of time

2) Body fat was estimated only by BCS. Despite the fact that BCS may correlate well with more accurate methods, this is a limitation of the study

3) Authors have mentioned other limitations of the study, including the number of dogs, the lack of information about their health status and lifestyle, and the fact that dogs belonged to a single breed

4) Potential correlation between low weight at birth and overweight in adulthood is nicely discussed but none of the mentioned hypotheses has been verified in the present study

5) In my opinion, Figure 1 may be deleted and BCS distribution of dogs may be reported in the text

Reviewer #2: The manuscript is well written and the results and conclusions are interesting. The overall presentation is sound.

Please add % behind the numbers in parenthesis in Table 1 "Neonatal and adult characteristics of overweight and not-overweight Labradors"

Please change "racial predisposition" in line 166 into "breed predisposition"

6. PLOS authors have the option to publish the peer review history of their article (what does this mean?). If published, this will include your full peer review and any attached files.

Reviewer #1: No

Reviewer #2: No

---

## [Author Response · Author response to Decision Letter 0]

18 Nov 2020

Journal requirements

• Please ensure that your manuscript meets PLOS ONE's style requirements, including those for file naming. The PLOS ONE style templates can be found at https://journals.plos.org/plosone/s/file?id=wjVg/PLOSOne_formatting_sample_main_body.pdf

and https://journals.plos.org/plosone/s/file?id=ba62/PLOSOne_formatting_sample_title_authors_affiliations.pdf

We have added the corresponding initials in brackets after the email address of the corresponding author and we have modified the layout of the first page. Next, we checked the overall style of the manuscript but we regret that we did not identify other areas of non-compliance with the guidelines. Could you be more specific about the problems identified?

• Thank you for stating in your Funding Statement: "This study was partially funded by Royal Canin SAS (Aimargues, France). AAL, JL

and JF participated in the analyses of the data and in the reviewing of the paper." Please provide an amended statement that declares *all* the funding or sources of support (whether external or internal to your organization) received during this study, as detailed online in our guide for authors at http://journals.plos.org/plosone/s/submit-now. Please also include the statement “There was no additional external funding received for this study.” in your updated Funding Statement. Please include your amended Funding Statement within your cover letter. We will change the online submission form on your behalf.

We have added the requested sentence at the Funding statement. 

• Thank you for stating the following in the Competing Interests section: "The authors have declared that no competing interests exist." We note that one or more of the authors have an affiliation to the commercial funders of this research study : Royal Canin.

Please provide an amended Funding Statement declaring this commercial affiliation, as well as a statement regarding the Role of Funders in your study. If the funding organization did not play a role in the study design, data collection and analysis, decision to publish, or preparation of the manuscript and only provided financial support in the form of authors' salaries and/or research materials, please review your statements relating to the author contributions, and ensure you have specifically and accurately indicated the role(s) that these authors had in your study. You can update author roles in the Author Contributions section of the online submission form. Please also include the following statement within your amended Funding Statement. “The funder provided support in the form of salaries for authors [insert relevant initials], but did not have any additional role in the study design, data collection and analysis, decision to publish, or preparation of the manuscript. The specific roles of these authors are articulated in the ‘author contributions’ section.”

We have provided this explanation and added the roles of authors affiliated to Royal Canin in Funding Statement.

Line 251 « This study was partially funded by Royal Canin SAS (Aimargues, France). The funder provided support in the form of salaries for AM. Moreover, AAL, JL and JF participated in the analyses of the data and in the reviewing of the paper. There was no additional external funding received for this study. »

Please also provide an updated Competing Interests Statement declaring this commercial affiliation along with any other relevant declarations relating to employment, consultancy, patents, products in development, or marketed products, etc. 

As noted earlier, there are no competing interests in this study. Royal Canin partially funded this study but that did not interfere with the full and objective publication of this research article.

Our commercial affiliation does not alter our adherence to PLOS ONE policies. So, we have added the requested sentence to the Competing Interests.

Line 246 « The authors have declared that no competing interests exist. The commercial affiliation does not interfere with the complete and objective presentation of this study, nor does it alter their adherence to PLOS ONE policies on sharing data and materiels. »

We have included the two updated sections in our cover letter. Thank you for proposing to make the changes online for us.

• Your ethics statement should only appear in the Methods section of your manuscript. If your ethics statement is written in any section besides the Methods, please move it to the Methods section and delete it from any other section. Please ensure that your ethics statement is included in your manuscript, as the ethics statement entered into the online submission form will not be published alongside your manuscript.

We have moved our Ethics approval to the Methods section.

 Line 116 : « Ethics approval

The protocol was reviewed and approved by the Royal Canin Committee for Animal Ethics and Welfare, reference 050917-39. »

Reviewer #1

This manuscript is about the association between birth weight and overweight in Labrador dogs.

The manuscript is well written, the statistical analysis seems appropriate to me as well as the conclusions, according to the results. However, the work has some limitations and needs some clarification.

• Since the age of the included dogs ranged from 1 to 9 years of age, it is assumed that this is a retrospective study. Authors are asked to explain how puppies were raised under similar environmental conditions over such a long period of time

We thank the reviewer for this question. All the puppies included were born in a professional breeding kennel with strict procedures in place, similar during the years under review (choice of food, frequency of weighing, cleaning of the environment…). This is why we wrote that « The study was conducted on Labradors raised under similar environmental conditions until two months of age which meant that the breed and breeding conditions were homogeneous. » (line 157). Even if, indeed, some environmental parameters such as hygrometry or temperature could not be controlled in the field and have probably varied over time.

• Body fat was estimated only by BCS. Despite the fact that BCS may correlate well with more accurate methods, this is a limitation of the study. Authors have mentioned other limitations of the study, including the number of dogs, the lack of information about their health status and lifestyle, and the fact that dogs belonged to a single breed

Thank you for this comment. We added the following sentence to clarify this limitation. 

Line 164 « Nevertheless, despite this good correlation, further studies with an assessment of adiposity by quantitative measures should be conducted. »

• Potential correlation between low weight at birth and overweight in adulthood is nicely discussed but none of the mentioned hypotheses has been verified in the present study

That’s true. This study is a first step in the investigation of the relationship between low birth weight and overweight in adulthood. As we mentioned in the discussion « further studies are needed to explore and identify the precise mechanisms explaining the association between low birth weight and adult overweight in the canine species » (line 204). 

• In my opinion, Figure 1 may be deleted and BCS distribution of dogs may be reported in the text

Thank you for this suggestion. We have chosen to converse the figure that provides a visual representation of the distribution of BCS in our population.

Reviewer #2

The manuscript is well written and the results and conclusions are interesting. The overall presentation is sound.

• Please add % behind the numbers in parenthesis in Table 1 "Neonatal and adult characteristics of overweight and not-overweight Labradors"

Thank you for this suggestion which clearly improves the readibility of the table. We have added %’s behind the figures in the table under consideration.

• Please change "racial predisposition" in line 166 into "breed predisposition"

Thank you for this suggestion. We modified this sentence.

Line 170 « In addition to breed predisposition, this may be explained by […] ».

---

## [Editor Report · Decision Letter 1]

27 Nov 2020

Association between birth weight and risk of overweight at adulthood in Labrador dogs

PONE-D-20-33064R1

Dear Dr. Mugnier,

We’re pleased to inform you that your manuscript has been judged scientifically suitable for publication and will be formally accepted for publication once it meets all outstanding technical requirements.

Kind regards,

Simon Clegg, PhD

Academic Editor

PLOS ONE

Additional Editor Comments:

Many thanks for resubmitting your manuscript to PLOS One

As you have addressed all the reviewer points, and the manuscript reads well, I have recommended it for publication

You should hear from the Editorial Office soon

It was a pleasure working with you and I wish you all the best for the future

Hope you are keeping safe and well in these difficult times

Thanks

Simon

---

## [Editor Report · Acceptance letter]

2 Dec 2020

PONE-D-20-33064R1 

Association between birth weight and risk of overweight at adulthood in Labrador dogs 

Dear Dr. Mugnier:

I'm pleased to inform you that your manuscript has been deemed suitable for publication in PLOS ONE. Congratulations! Your manuscript is now with our production department. 

Kind regards, 

on behalf of

Dr. Simon Clegg 

Academic Editor

PLOS ONE